# $F_{ST}$ and genetic diversity in an island model with background selection

**Asad Hasan** *, **Michael C. Whitlock**

Department of Zoology, University of British Columbia, Vancouver, British Columbia, Canada

* a.hasan@zoology.ubc.ca

## Abstract

Background selection, by which selection on deleterious alleles reduces diversity at linked neutral sites, influences patterns of total neutral diversity, $\pi_T$, and genetic differentiation, $F_{ST}$, in structured populations. The theory of background selection may be split into two regimes: the *background selection regime*, where selection pressures are strong and mutation rates are sufficiently low such that deleterious alleles are at a deterministic mutation-selection balance, and the *interference selection regime*, where selection pressures are weak and mutation rates are sufficiently high that deleterious alleles accumulate and interfere with another, leading to selective interference. Previous work has quantified the effects of background selection on $\pi_T$ and $F_{ST}$ only for deleterious alleles in the *background selection regime*. Furthermore, there is evidence to suggest that migration reduces the effects of background selection on $F_{ST}$, but this has not been fully explained. Here, we derive novel theory to predict the effects of migration on background selection experienced by a subpopulation and extend previous theory from the *interference selection regime* to make predictions in an island model. Using simulations, we show that this theory best predicts $F_{ST}$ and $\pi_T$. Moreover, we demonstrate that background selection may generate minimal increases in $F_{ST}$ under sufficiently high migration rates, because migration reduces correlated effects on fitness over generations within subpopulations. However, we show that background selection may still cause substantial reductions in $\pi_T$, particularly for metapopulations with a larger effective population size. Our work further extends the theory of background selection into structured populations, and suggests that background selection will minimally confound locus-to-locus $F_{ST}$ scans.

## Author summary

Most mutations that affect fitness incur deleterious effects and are ultimately removed via natural selection. Consequently, nearby neutral variants may also experience the effects of selection; this is termed background selection. Background selection greatly influences patterns of genetic diversity both between and within populations among virtually all extant species, and is therefore of great interest to geneticists. Previous models of background selection have been primarily restricted to populations with completely random mating. However, it is well known that most natural populations exhibit some form of

**Data Availability Statement:** All of the data from our simulations and code for our analyses of our simulations, computation of B and Blocal, and plotting is available on GitHub (https://github.com/asadrh8/FST-BGS).

**Funding:** MCW was funded by an NSERC (Natural Sciences and Engineering Research Council of Canada) Discovery Grant (Grant number: RGPIN-2016-03779). The funders had no role in study design, data collection and analysis, decision to publish, or preparation of the manuscript.

**Competing interests:** The authors have declared that no competing interests exist.

spatial structure. Here, we explore the effects of background selection in spatially structured populations, and we find that migration between subpopulations may attenuate the effects of background selection acting to increase genetic differentiation among populations. We derive novel theory to account for this effect by considering that individuals with deleterious alleles may migrate out of a local subpopulation prior to being removed by the population via selection. Our work demonstrates that, when migration rates are high, background selection does not substantially influence genetic differentiation among populations. Despite this, we find that background selection may greatly decrease genetic diversity within subpopulations and in the whole metapopulation.

## Introduction

Background selection is the process by which selection on deleterious alleles also affects diversity at linked sites [1–2]. Background selection reduces genetic diversity, as individuals carrying deleterious mutations are selected out of the population, and ultimately do not contribute to neutral diversity [1,3–4]. Due to the ubiquity of deleterious mutations [5–6], background selection occurs in all species and its effects have been estimated in several species, including humans [7–11], fruit flies [12–14], and various other eukaryotes [15]. Given that background selection influences patterns of genetic variation and the site frequency spectrum [16–18], it may bias demographic inference or inferences of other selective processes [14,19]. Thus, there is a general recognition of the effects of background selection on patterns of diversity, and efforts are underway to incorporate background selection into null models of genomic diversity [9,13,14,20].

The strength of background selection is approximated using the metric $B$, which can be defined as the ratio of the effective population size under background selection and the effective population size without background selection. In other words $B = \frac{N_e}{N_{e0}}$, where $N_e$ is the effective population size with background selection, and $N_{e0}$ is the effective population size without background selection. With the correct effective population size, $B$ predicts the genetic diversity ($\pi$) relative to that expected without background selection ($\pi_0 \approx 4N_{e0}\mu$ in a diploid population under an infinite sites model [21], where $\mu$ is the neutral per base-pair mutation rate). It is important to note that because this approximation simply rescales the effective population size, it does not fully capture all the features of background selection. In particular, it does not capture the influence of background selection on the site frequency spectrum due to distortions in genealogies [2,18,22–23]. However, this approximation works reasonably well under most conditions [18].

Generally, we can think of purifying selection as introducing positive correlations between generations in deleterious allele frequency change, amplifying the variance in reproductive success (and therefore genetic drift) experienced by linked neutral sites, further reducing their $N_e$ beyond a simple mutation-drift model [4,24–26]. Individuals with deleterious mutations are less likely to have offspring, and so are their offspring as they may also carry deleterious mutations. $B$ therefore captures the amplification of variance in reproductive success of neutral alleles due to their multigenerational association with deleterious alleles.

Most natural populations exhibit some degree of population structure [27], and this influences patterns of genetic diversity. In structured populations, $F_{ST}$ [28] is a widely used population genetic metric among plant/animal breeders, conservationists, and others [29] to identify sites subject to spatially varying selection (in the case of $F_{ST}$ outlier tests), the demographic histories of subpopulations, rates of local inbreeding, and for various other reasons. $F_{ST}$ has

various definitions. It can be defined as $F_{ST} = 1 - \frac{\pi_S}{\pi_B}$, where $\pi_S$ is the average proportion of pairwise differences between haplotypes within a subpopulation and $\pi_B$ is the average proportion of pairwise differences between haplotypes from different subpopulations [30; the same quantity is estimated by ref. [31] $\theta$; see ref. [32]. $F_{ST}$ can also be defined in terms of the standardized variance in allele frequencies among subpopulations, or in terms of mean coalescence times for alleles within subpopulations versus the whole population [33].

Background selection influences neutral diversity patterns in populations, [34–37], and this effect varies across the genome. Variation in recombination rates, mutation rates, and the density of functionally conserved regions generate locus-to-locus variation in $N_e$ and $F_{ST}$, which has been observed in human [8] and fruit fly [32] populations. Locally beneficial alleles are expected to exhibit greater $F_{ST}$ values relative to neutrality, as they exhibit greater variance in frequency between subpopulations relative to neutral alleles. However, locus-to-locus variation in background selection can increase $F_{ST}$ values for neutral loci as well, confounding scans for locally adapted alleles and potentially increasing the false-positive rate. Work by Matthey-Doret and Whitlock has shown this effect to be weak, particularly when mean $F_{ST}$ values are low [37]. This work also suggested an effect of migration to increase $B$ (reduce the effect of background selection on $N_e$ and $\pi$). In other words, when migration rates are sufficiently high, it is possible some of the effects of background selection are not realized in a local population before the allele is lost to selection, potentially weakening the confounding effect of background selection on $F_{ST}$. However, this effect remains analytically unexplored. The central objective of this paper is therefore to test and improve the approximations used to predict the effects of background selection on $F_{ST}$ and total diversity, $\pi_T$, of a metapopulation.

Classical models of background selection estimated $B$ under the assumption that the effective strength of selection was not weak ($|N_e t| > 1$, where $t$ is the heterozygous selection coefficient in diploids) and mutations are sufficiently rare, such that the average frequency of deleterious alleles is well predicted by a deterministic mutation-selection balance [1,3–4]. This parameter range has become known as the *background selection regime*. In the *background selection regime*, the dynamics at deleterious loci can be assumed to be independent from one another, and genetic drift can be ignored. Background selection was first modeled assuming an infinitely large diploid population where deleterious loci are at mutation-selection balance in a non-recombining genome. In this case the fraction of deleterious mutation-free gametes could be modeled using a Poisson distribution with mean $\frac{U}{t}$, where $U$ is the gametic genomic deleterious mutation rate [1]. Soon after, this model was extended to consider recombination [3–4]. From the very beginning [1], these papers acknowledged that background selection with weaker selection would behave differently; in particular weak selection was shown to have differing effects on the number of segregating sites [38].

Later work extended our understanding of cases where selection against deleterious alleles is weak (i.e., $|N_e t| < 1$), recombination rates approach zero ($r \to 0$), and the per base-pair mutation rate, $\mu$, is sufficiently high such that deleterious alleles are introduced frequently and inefficiently removed by selection [11,23,26,39–40]. Here, weakly deleterious alleles accumulate along the genome, and the effects of selection at a single site are influenced by selection at other linked sites, leading to the "Hill-Robertson effect" or selective interference [40–41]. In this *interference selection regime*, assumptions of a deterministic mutation-selection balance fail to predict the quantitative effects of background selection, as deleterious alleles ($|N_e t| < 1$) exhibit stochastic fluctuations in frequency due to the weak efficacy of selection and correlations between deleterious sites. Contrary to theory from the *background selection regime*, models of selective interference cannot assume independence among selected sites and must consider multiple selected sites in tight linkage with one another [39,40] as well as the effects

of genetic drift [42]. There has been some work to address this issue, by instead considering the total variance in fitness experienced by a neutral allele under background selection in a region under selection, circumventing the issue of disentangling individual selection pressures. For instance, it has been shown that the effects of many weakly deleterious alleles in this *interference selection regime* can be approximated by fewer, strongly deleterious alleles in the *background selection regime* that generate equal variance in fitness [23]. Similarly, other work has shown that *B* can be approximated for weakly deleterious alleles by considering their fixation at an appreciable rate by genetic drift, as we expect substituted mutations to no longer contribute to fitness variance [11,26] These models effectively circumvent the issue of selective interference to predict *B* in panmictic populations for both the *background selection* and *interference selection regimes* (see refs.[2, 43] for reviews on background selection). However, we still lack theory to make predictions about patterns of genetic diversity in structured populations under background selection using these models, and they have not been extended to consider the potential effects of migration.

In models of structured populations under background selection, the equilibrium effects of background selection on $F_{ST}$ have been approximated by rescaling the local population size of a subpopulation by *B* [36]. This is intuitive, as *B* represents the effects of background selection on the rate of genetic drift on neutral loci, with smaller values of *B* corresponding to greater amounts of genetic drift; thus, a greater effect of background selection on neutral loci will lead to a reduced local effective population size, $N_{e,local}$. In a haploid two-island model, Zeng and Corcoran [36] predicted that

$$F_{ST} \approx \frac{1}{4BN_{local}m + 1},$$

(1)

where $N_{local}$ is the local population size, and *m* is the total migration rate. Note that $BN_{local} = N_{e, local}$, and that this equation is consistent with the more general diploid finite island model prediction of $F_{ST} \approx \frac{1}{4N_e m \frac{d}{d-1} + 1}$, where *d* is the number of demes [32]. This approximation has been shown to work well to predict $F_{ST}$ in the deterministic *background selection regime*, in cases with strong selection and low migration rates [36]. However, no previous study provides a solution to the effects of background selection on $F_{ST}$ for weakly deleterious alleles in the *interference selection regime*. This represents a gap in the literature, as both the theory [44] and empirical estimates of the distribution of fitness effects (reviewed in Ref. [5]) suggest that a non-trivial portion of coding and non-coding mutations incur weakly deleterious effects on fitness (see also ref.[6]). Furthermore, we lack theory predicting the effects of migration on background selection or the effects of background selection on total diversity, $\pi_T$, in structured populations. Given that spatial structure is a common feature of natural populations, we require theory to better account for its role in shaping $\pi_T$.

The overall goal of this paper is therefore to better understand the effects of background selection in structured populations. We find that some previous approximations fail under common biological conditions, and we develop new approximate theory. We use simulations to test and compare the accuracy of these approximate theories in predicting $\pi_T$ and $F_{ST}$ under background selection in an island model. Our simulations show that the effects of background selection from weakly deleterious alleles (in the *interference selection regime*) on $F_{ST}$ are relatively weak, and that the effects of background selection on $F_{ST}$ are attenuated by migration; with high migration, $F_{ST}$ values approach those expected under neutrality. We derive novel background selection theory to account for this migration effect. We also combine theory about background selection from the *interference selection regime* in panmictic populations [11,23,26] with what is known about how background selection affects $F_{ST}$ [36] to best estimate

$F_{ST}$ and $\pi_T$ in an island model for both the *interference selection* and *background selection regimes* under various strengths of selection, recombination rates, and migration rates. Lastly, we demonstrate that, while the effects of background selection from weakly deleterious alleles on $F_{ST}$ are weak, background selection from weakly deleterious alleles can substantially reduce total $\pi_T$ in a large metapopulation, and that we can predict this effect reasonably well. Our work extends the theory of background selection from panmictic populations to structured populations and lends further support for an insignificant confounding role of background selection in $F_{ST}$ scans for populations where $F_{ST}$ is relatively low on average. Our results also suggest a potentially under-appreciated role for background selection from weakly deleterious alleles in reducing total diversity in sizable metapopulations.

## Theory

Zeng and Corcoran [36] suggested that the effects of $F_{ST}$ in an island model would be predictable from modifying the effective size of a subpopulation, $N_{local}$, by multiplying times $B$, the $N_e/N$ ratio created by background selection. We show here that migration lessens the effects of background selection on the amount of genetic differentiation caused by local drift relative to predictions of $B$ based on earlier work (e.g. Ref [4]). Then, we derive the expected $F_{ST}$ in an island model with background selection and a finite number of demes, when migration follows genetic drift.

## Derivation of $B$ with 'migration effect'

In this section, we extend the quantitative genetic approach to modeling background selection, following the framework of Santiago and Caballero [24–26] and Buffalo and Kern [11], to account for how migration influences the effects of background selection, $B$, in an island model consisting of diploid individuals. In brief, these models consider how the additive genetic fitness variance experienced by a neutral allele due to its associations with deleterious alleles further increase its variance in reproductive success, therefore reducing its effective population size beyond that of a mutation-drift model. Given that this additional increase in the variance of reproductive success of neutral alleles is due to selection at linked sites, this has been termed the *fitness*-effective population size, $N_f$ [11]. First, we describe the framework under panmixia; then, we extend it to predict the geographically local effects of background selection in an island model, $B_{local}$ (where $B_{local} = \frac{N_{f,local}}{N_{local}}$), by deriving its local fitness-effective population size $N_{f,\ local}$. [24,45].

In a panmictic population, Santiago & Caballero [25–26] suggest that, in order to understand the effects of background selection on $N_f$ at a neutral allele, we must consider both the magnitude of additive genetic fitness variance created by associated deleterious alleles, $V_A$, as well as the cumulative effects of this variance on the reproductive success of a neutral allele over time, estimated by the linkage inflation factor $Q^2$. The product of these two, $V_A Q^2$, therefore represents the total variance in reproductive success over time experienced by a neutral allele due to the effects of purifying selection at linked sites. This is summarized by the following equation:

$$N_f = \frac{N}{1 + V_A Q^2}$$

Assuming a multiplicative polygenic basis to fitness, we can approximate $N_f$ at a neutral site linked to $S$ regions under selection as $N_f \approx N \exp\left(-\sum_{i=1}^{S} V_{A,i} \frac{Q_i^2}{2}\right)$. Here, we follow Buffalo and

Kern [11] and ignore short-lived associations of neutral sites with sites on homologous chromosomes, hence the division by 2.

For our derivation, we now consider an island model with $d$ subpopulations, each with $N_{local}$ diploid individuals. Deleterious mutations appear at site $i$ with per-base pair mutation rate $\mu_i$ and heterozygous selection coefficient $t_i$. Recombination between the focal neutral site and deleterious allele $i$ occurs at rate $r_i$. In order to predict the effects of background selection on $F_{ST}$ in an island model, we now consider the effects of background selection on the fitness-effective population size of each geographically local subpopulation, $N_{f, local}$.

First, we will solve for $Q_i^2$, the linkage inflation factor. $Q_i^2$ captures the cross-generational associations between a focal neutral site and deleterious site $i$ within a focal subpopulation. For now, we will consider the case of strong selection; the estimation of $Q_i^2$ is later extended to address weak selection. The magnitude of $Q_i^2$ is contingent on the heterozygous selection coefficient of the deleterious allele, $t_i$, and the recombination rate between the neutral and deleterious allele, $r_i$. This is intuitive, as we expect the association between a neutral and deleterious allele within a focal subpopulation to be ended if the individual is selected out of the population or if the neutral allele recombines off of the background of the deleterious allele. Here, we account for the fact that the association between a neutral and deleterious allele also ends within a focal subpopulation if the individual carrying the haplotype emigrates out of the subpopulation, which will occur with rate $m$. Thus, the autocorrelation function for a neutral allele associated with deleterious allele $i$ over time is $C(\tau) = [(1 - r_i)(1 - t_i)(1 - m)]^\tau$, where $\tau$ is the number of generations. In order to estimate the net effect of this autocorrelation, $Q_{\infty,i}^2$, we sum its effects over time:

$$Q_{\infty,i}^2 = \left[\sum_{\tau=0}^{\infty} C(\tau)\right]^2 = \left[\frac{1}{1 - (1 - r_i)(1 - t_i)(1 - m)}\right]^2.$$

Next, we turn to solving $V_{A,i}$, the additive genetic variance contributed by deleterious alleles in segment $i$. Here, we simply follow the results of Buffalo and Kern [11], who show that $V_{A,i}$ is well approximated in a region of $L_i$ selected sites each with heterozygous selection coefficient $t_i$ and mutation rate $\mu_i$ under mutation-selection balance to be $V_{A,i} \approx U_i t$, where $U_i = 2L_i\mu_i$ is the total mutation rate in the segment. Thus, for $S$ selected segments, we get the following equation for $N_{f,local}$:

$$N_{f,local} \approx N_{local}\exp\left[-\sum_{i=1}^{S} \frac{\mu_i L_i}{\left(1 - (1 - r_i)(1 - t_i)(1 - m)\right)^2}\right].$$

As pointed out by previous work, classic background selection theory assuming mutation-selection balance only serves as a reasonable approximation under strong selection and/or weak mutation (see ref. [23]). When alleles are weakly deleterious, we expect genetic drift to substantially affect their frequency dynamics such that they are no longer well predicted by classic mutation-selection balance theory. Santiago and Caballero [25–26] demonstrated that $N_f$ can be well approximated under weak selection by considering that some fraction of weakly deleterious alleles fix due to genetic drift, thus no longer contributing to the additive genetic variance, $V_A$, experienced by linked neutral alleles.

Under this weak selection regime in diploids, where deleterious alleles have a non-trivial rate of fixation, $V_A$ may be approximated as $V_A = (U-2R)t$ [11], where $R$ is the rate of fixation of deleterious alleles (the factor of two is due to our modeling of diploids). $R$ is estimated by considering the probability of fixation of deleterious alleles [46], $p_{fix}$, as well as their rate of introduction to the population, $NU$, the product of the population size and diploid genomic

deleterious mutation rate. Thus, we get the following system of non-linear equations, which can be numerically solved for $N_{f,local}$ and $R$:

$$N_{f,local} = N_{local}B_{local} \approx N_{local}\exp[-\sum_{i=1}^{S}(U-2R)t\frac{Q_i^2}{2}] \qquad (2)$$

$$R_{local} = \frac{4N_{f,local}Ut}{e^{4N_{f,local}t}-1}. \qquad (3)$$

Under weak selection, we must also modify $Q^2$ to account for the loss of additive genetic variance [11, 26]. Under the strong selection regime, it was assumed that the rate of decay in association between a focal neutral allele and linked deleterious alleles due to reductions in fitness variance from selection was equal to the heterozygous selection coefficient $t$. This is derived by considering that the factor estimating the decay in fitness variance due to selection, $Z$, is estimated as $Z = 1 - \frac{V_M}{V_A}$, where $V_M$ is the mutational variance. A more in-depth explanation is provided in the supplemental materials of Buffalo and Kern [11]. Briefly, Buffalo and Kern [11] demonstrate that $V_M \approx Ut^2$ and $V_A = (U-2R)t$. In the case of strong selection, where $R \cong 0$, we recover $Z = 1 - \frac{V_M}{V_A} = 1 - t$, matching the term described in the original, strong selection derivation of $Q^2$ above. Under weak selection, where $R > 0$, we instead get $Z = 1 - \frac{Ut}{U-2R}$. Thus, more generally, $Q_{\infty,i}^2 = [\sum_{\tau=0}^{\infty}[Z_i(1-r_i)(1-m)]^{\tau}]^2 = [\frac{1}{1-Z_i(1-r_i)(1-m)}]^2$. Buffalo & Kern [11] use a continuous approximation to derive $Q^2$ over a segment under selection of map length $M_i$. We also follow this method with addition of our $(1-m)$ term, finding that

$$Q_{\infty,i}^2 \approx \frac{2}{M_i}\int_0^{M_i/2}Q_{\infty,i}^2(r)dr = \frac{2}{(1-Z_i-Z_im)(2-Z_i(2-M_i)+Z_im(2-M_i))}$$

(*cf*. ref.[11], supplement Eq 33), where $M_i$ is the map length in morgans. This derivation relies on a few assumptions: $r_i$ is small such that it is additive over sites, $t_i$ is constant over the region, and the neutral site is embedded in the center of the region.

For our predictions of $\pi$ and $F_{ST}$ in an island model, we numerically solve for $N_{f,local}$ and $R_{local}$ for our parameters of interest to get $B_{local}$ ($B_{local} = \frac{N_{f,local}}{N_{local}}$), thus allowing us to make predictions across both weak and strong deleterious selection coefficients. One caveat to this model, however, is the lack of an explicit treatment of the effects of interference. Under weak selection and low recombination, we expect correlations in frequency to arise between segregating deleterious variation, leading to selective interference among deleterious alleles. Here, selection at a deleterious site is no longer independent of selection at nearby sites. Instead, deleterious sites themselves also experience a reduction in fitness-effective population size due to selection at linked sites. To account for this, we follow the methodology of Buffalo & Kern [11] and rescale $N_{local}$ at a region $x$ as $N_{local,x} = B_{local,x}N$. We then use this re-scaled $N_{local,x}$ instead of $N_{local}$ to numerically solve for $N_{f,local}$ and $R$.

Santiago and Caballero [26] showed that the effective population size for calculating fixation rates ("fixation effective size", approximated by $N_f$) with background selection is different from the effective population size for predicting neutral diversity ("heterozygosity effective size", $N_{eH}$). To predict $\pi$, we must slightly modify the above theory to instead estimate $N_{eH}$. We may do so by considering that it is not the asymptotic linkage inflation factor, $Q_{\infty}^2$, that is most relevant to prediction of metrics of genetic diversity [11 supp., 26] Instead, at equilibrium, there is variation in the time in which neutral alleles have arisen in the past and therefore the degree to which they have experienced the effect of background selection. Thus, we expect $N_{eH}$

to be less than $N_{local}$ but greater than $N_f$, the effective population size estimated from using $Q_\infty^2$. More details may be found in the *Methods*.

## Prediction of $F_{ST}$

Previous work on background selection in structured populations has demonstrated that, in some cases, the effects of background selection in an island model can be approximated as a reduction in local effective population sizes proportional to the strength of background selection [34, 36]. In order to estimate and predict $F_{ST}$, we use the definition of Hudson et al. [30], where

$$F_{ST} = 1 - \frac{\pi_S}{\pi_B}.$$

We show in Eq A4 in S1 Appendix that, using the neutral expectation of $F_{ST}$ in an island model, predicted $F_{ST}$ under background selection is:

$$F_{ST,predicted} = \frac{1}{1 + \frac{(d-1)d(4-2m)N_{local}B_{local}m}{(d(1-m)-1)^2}} \tag{4}$$

Here, $m$ is the total proportion of immigrants into a deme per generation, $d$ is the number of demes, and $B_{local}$ is a measure of background selection experienced by the subpopulation. Our equation differs from that of Ref. [32] because, in SLiM (the evolutionary simulation framework used; ref. [47]), the order of events is such that offspring production occurs first (genetic drift) followed by migration and then $F_{ST}$ estimation [48], and because we consider terms that are $m^2$ or higher.

We note that there are multiple definitions of $F_{ST}$ that have been previously used. For instance, the equation used to estimate $F_{ST}$ in our simulations uses the ratio of $\pi_S$ and $\pi_B$, but an alternative definition uses the ratio of $\pi_S$ and $\pi_T$, the total population diversity. This is equivalent to $G_{ST}$, and is defined as:

$$G_{ST} = 1 - \frac{\pi_S}{\pi_T}$$

[49]. Using the neutral expectation, $G_{ST}$ in a finite island model experiencing background selection with migration following drift and then $F_{ST}$ estimation is derived in Eq A5 in S1 Appendix to be:

$$G_{ST,predicted} = \frac{1}{1 + \frac{d^2(4-2m)N_{local}B_{local}m}{(d(1-m)-1)^2}}. \tag{5}$$

## Results

We use a simulation-based approach to investigate the accuracy of our theoretical derivation of background selection in the *interference selection* and *background selection regimes* under varying migration rates. In SLiM v3.7.1 [47], we run simulations of a metapopulation using an island model consisting of $N_{global}$ diploid individuals split equally into $d$ demes. Population size is held constant, and generations are non-overlapping. Each generation, a total fraction of $m$ individuals in each deme are replaced by immigrants, equally partitioned from each other deme. We simulate a genomic region composed of $L_{neutral}$ neutral sites, surrounded on each side by $L_{selected}/2$ selected sites for a total of $L_{selected}$ sites subject to deleterious mutation.

Recombination occurs between the neutral region and selected sites and between selected sites at total rate $M$ (denoting the map length). Deleterious mutations are introduced to gametes at fixed rate $U$, with fixed selection coefficients against heterozygotes $t$. Fitness is multiplicative across sites and additive within sites. More details are given in the *Methods* section.

## Background selection in structured populations and prediction by previous models

For our high migration cases (Fig 1A and 1B), $F_{ST}$ under neutrality was 0.004, representing populations with extremely low genetic differentiation and high gene flow. For our intermediate migration cases (Fig 1C and 1D), $F_{ST}$ under neutrality was ~0.05, representing populations with moderate genetic differentiation. For our low migration cases (Fig 1E and 1F), $F_{ST}$ under neutrality was ~0.50, thus representing populations with very high genetic differentiation and low gene flow. Hereafter, we consider deleterious alleles with $N_{local}t \leq 1$ to be weakly deleterious, $1 < N_{local}t < 7.5$ to be moderately deleterious, and $N_{local}t \geq 7.5$ to be strongly deleterious. Given previous work on the theory of background selection in structured populations [36], we expected the greatest increase in $F_{ST}$ due to background selection at moderate strengths of selection. This is intuitive, as we expect for strongly deleterious mutations to be rapidly purged from the population, thus generating weak total variance in fitness due to their short duration in the population. Weakly deleterious mutations, on the other hand, may persist for many more generations in the population, but simply generate little total variance in fitness due to their individually weak selective effects. Moderately deleterious alleles, however, generate moderate variance in fitness and are not rapidly purged from the population; thus, we expected an intermediate maximum of $F_{ST}$ with respect to strength of selection against deleterious alleles–in other words, an inverted U-shaped curve.

Indeed, we observed an inverted U-shape curve of $F_{ST}$ under background selection, but only for our low migration cases (Fig 1E and 1F), similar to the effect on overall diversity from background selection [50]. We later discuss the intermediate and high migration cases. Considering only our low migration cases, we observed that the effects of background selection on $F_{ST}$ from weakly deleterious alleles are relatively weak (Fig 1E and 1F, gray dots when $N_{local}t \leq 1$). As expected, we also observed that the effect of background selection on $F_{ST}$ is attenuated by recombination (*cf.* Fig 1E to 1F); recombination introduces the opportunity for a neutral allele to recombine off of the background of a deleterious allele, thus rescuing it from eventual loss via linked selection.

As expected, classic theory from the *background selection regime* (refs. [1,3–4]; Fig 1, purple dots) substantially overestimates the effects of background selection (underestimates $B$) for weakly deleterious alleles (see ref. [23]) and therefore overestimates the effects on $F_{ST}$ [36], with the discrepancy increasing for weaker strengths of selection (Fig 1 insets, *cf.* purple to gray dots). As $t$ approaches 0, the classical formula for $B$ inaccurately predicts deleterious allele frequencies, and indeed those papers [1,3–4] deriving the predictions are clear that the derivation applies to stronger effect mutations. The discrepancy between observation and prediction, however, is somewhat attenuated in the $M = 0.01$ cases (*cf.* Fig 1E inset to Fig 1F inset), presumably due to recombination reducing the magnitude of selective interference. Nevertheless, classic theory poorly predicts the effects of background selection on $F_{ST}$ for weakly deleterious alleles, as expected.

On the other hand, our novel derivation of the effects of background selection (Fig 1, pink dots) estimates $F_{ST}$ well when alleles are weakly deleterious. However, the theory does not accurately predict $F_{ST}$ when $N_{local}t \approx 2.5$ (Fig 1E and 1F). Our inability to predict $B$ in this regime previously been pointed out by Buffalo and Kern [11], and the accurate estimation of $B$

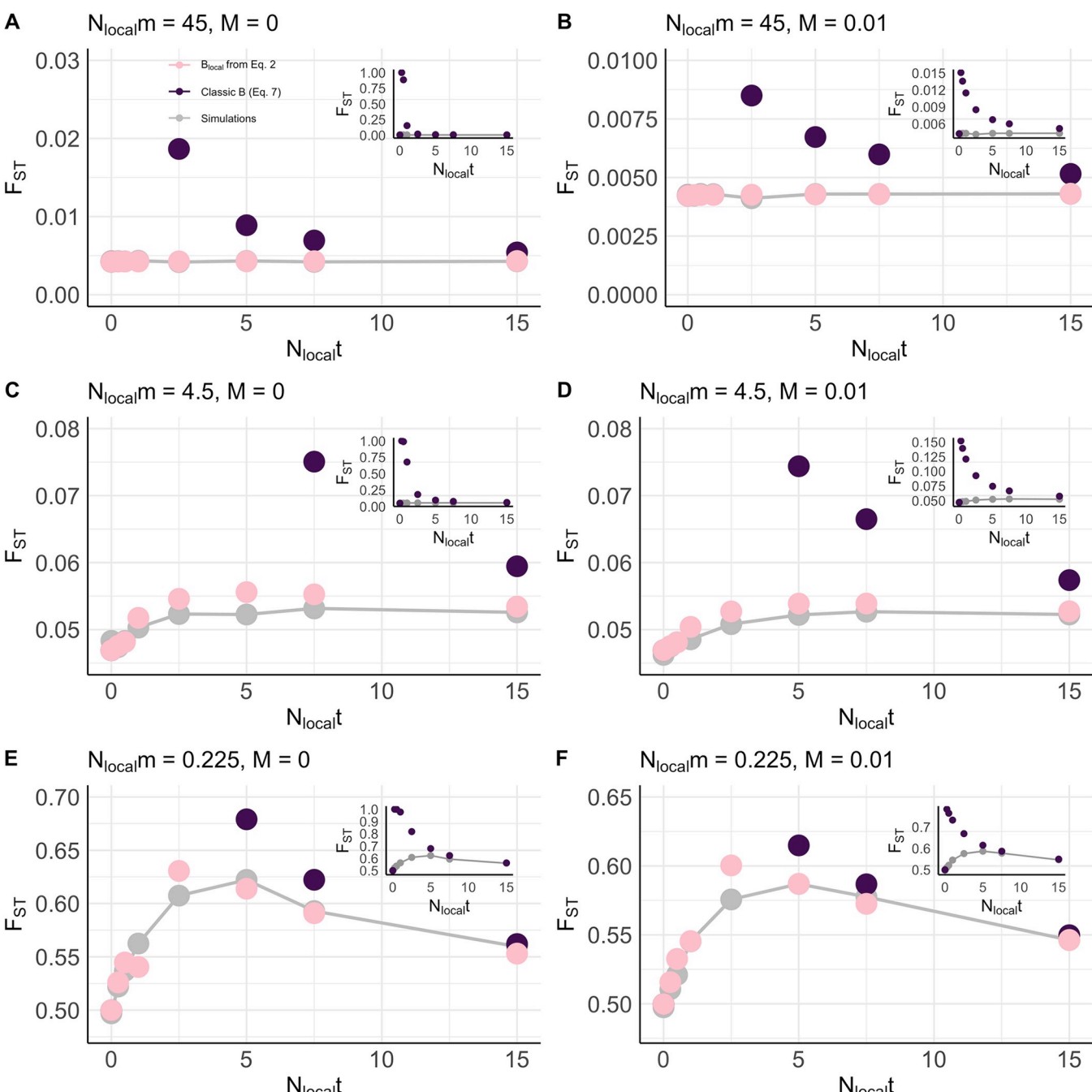

**Fig 1. Accuracy of various methods in predicting $F_{ST}$ with respect to the scaled heterozygous strength of selection, $N_{local}t$.** Forward-time simulations run with (A) $N_{local}m = 45$ and $M = 0$, (B) $N_{local}m = 45$ and $M = 0.01$, (C) $N_{local}m = 4.5$ and $M = 0$, (D) $N_{local}m = 4.5$ and $M = 0$, (E) $N_{local}m = 0.225$ and $M = 0$, and (F) $N_{local}m = 0.225$ and $M = 0.01$. The gray dots connected by a line represent $F_{ST}$ values from forward-time simulations, with 95% CIs. (These were quite small, and not always visible.) $F_{ST}$ was calculated using Eq 4, which requires a value for $B_{local}$. The pink dots represent theoretical predictions of $B_{local}$ from our novel derivation accounting for the effects of migration (Eq 2) with re-scaling of $N_{local}$ to approximate the effects of selective interference, and the purple dots represent theoretical predictions of $B_{local}$ from classic background selection theory (Eq 7 in our Method section; refs. [4, 36]). The insets show the performance of classic background selection theory to structured populations following ref. [36] for all simulated $N_e t$. All parameter combinations were simulated 750 times in a 10-deme island model with local population size $N_{local} = 500$, locally scaled selection coefficients $N_{local}t = \{0, 0.25, 0.5, 1, 2.5, 5, 7.5, 15\}$, $N_{local}m = \{45, 4.5, 0.225\}$, $U = 7 \times 10^{-3}$, $L_{selected} = 700$, and for 250,000 generations. In many cases (especially in part D), the plot points for the simulations (gray dots) are hidden behind the points shown for theoretical expectations. Note the different scales on the y-axis between plots.

when $2N_e t \approx 1$ remains an outstanding problem in the background selection literature. Thus, we expect some inaccuracy from the theory here.

Next, we turn to our case of high migration rates. Here, we observe no effects of background selection on $F_{ST}$ for all simulated values of $N_{local}t$ (Fig 1A and 1B). This lack of effect of background selection on $F_{ST}$ under higher rates of migration has been observed previously [37]. However, classic background selection theory still predicts an exponential decline in $F_{ST}$ with respect to $N_{local}t$ (Fig 1A and 1B, purple dots). Additionally, note that background selection theory with consideration of interference but not migration (following the method of ref. [23]) erroneously predicts an inverted U-shaped curve with respect to $N_{local}t$ (see Fig A in *S1 Text* green dots) under high migration rates.

In our intermediate migration case (Fig 1C and 1D), we observe some elevation of $F_{ST}$ at moderate strengths of selection, although quite weak relative to our low migration case due to the effects of moderate levels of migration (*cf.* Fig 1C and 1D to Fig 1E and 1F). Our theoretical derivation of $B_{local}$ is also able to predict this effect well. In our derivation above, we hypothesized that migration acts to reduce the effect of background selection by presenting a neutral allele the opportunity to end its association with deleterious alleles within a subpopulation, thus reducing its total variance in fitness within the focal subpopulation. We further discuss these predictions below.

## 'Migration effect' theory accurately predicts the effects of background selection on $F_{ST}$

The application of our derivation of $B_{local}$ with the 'migration effect' (Fig 1, pink dots) accurately predicts $F_{ST}$ under various fixed strengths of selection, recombination rates (Fig 1), and migration rates (Fig 2). Moreover, we find that this accuracy extends well into the *interference selection regime*, predicting $F_{ST}$ well under lower strengths of selection. As expected, the 'migration effect' is particularly prominent at higher migration rates. Under our higher migration rate parameter set (Fig 1A and 1B), we observed no effect of background selection on $F_{ST}$, despite marked reductions in $\pi_T$ (Fig 3A and 3B). Thus, migration appears to attenuate the effects of background selection by exporting deleterious alleles to different subpopulations prior to their removal via selection, resulting in similar reductions in $\pi_T$ and $\pi_S$ such that $F_{ST}$ is not elevated. In contrast, with low migration, background selection occurs primarily within subpopulations to reduce $\pi_S$, with some effect on $\pi_T$ [35,37].

We explored the attenuating effect of background selection on $F_{ST}$ across multiple migration rates (Fig 2). We find that incorporation of the migration effect to estimate the geographically local effects of background selection, $B_{local}$, predicts $F_{ST}$ more accurately than our estimates without incorporation of the migration effect (*cf.* Fig 2 pink & turquoise dots). For migration rates of $m = 0.01$ and higher, we also find that $F_{ST}$ values strongly resemble $F_{ST}$ values expected under no background selection (Fig 2 inset, *cf.* gray to dark blue dots). Thus, our results suggest that the alignment of observed $F_{ST}$ value to neutral prediction at low $F_{ST}$ values may not be sufficient to predict the absence of background selection in empirical systems, as it may otherwise be explained by the presence of both background selection and high migration rates. Regardless, our theoretical predictions further support the expectation of background selection weakly confounding locus-to-locus $F_{ST}$ scans when gene flow is sufficiently common [37]. Our simulations suggest that this is true in an island model when $m \geq t$ (Fig 2 inset).

## Background selection may substantially reduce $\pi_T$

In this section we highlight some general effects of background selection on genetic diversity, $\pi$, in our simulations and discuss the performance of our theoretical predictions. We found

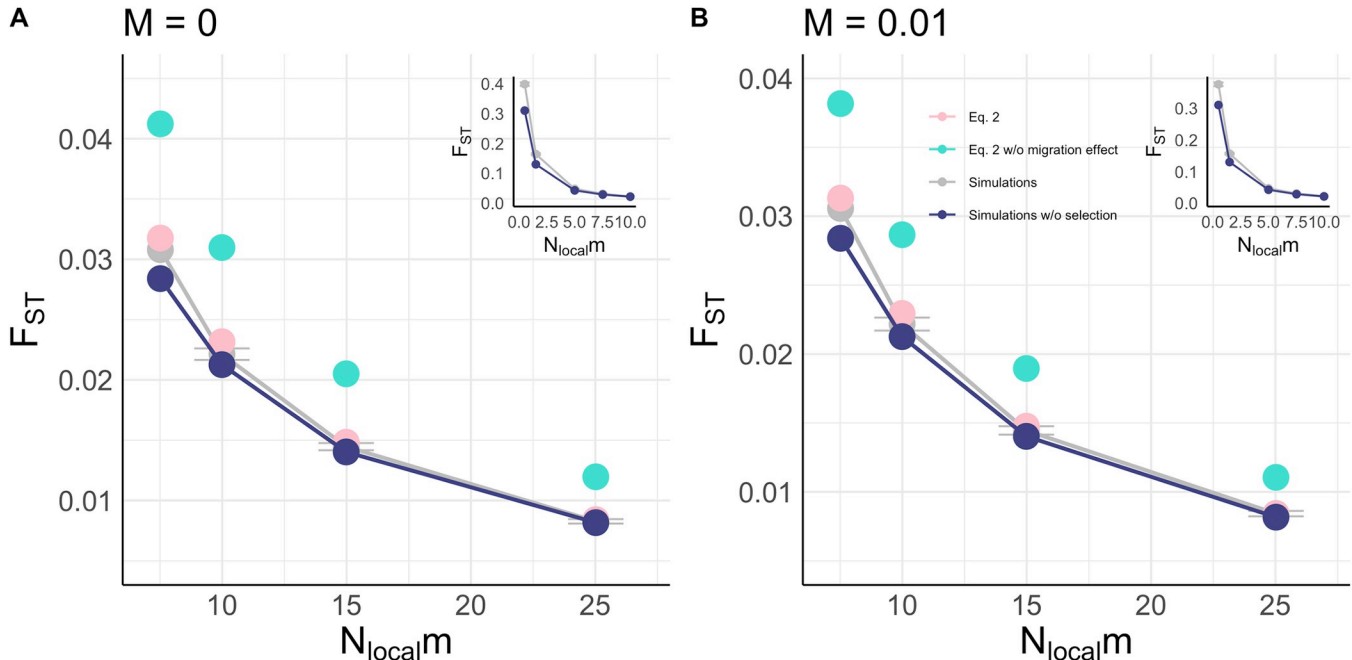

**Fig 2. Inclusion and exclusion of migration effect in estimation of $F_{ST}$ under background selection.** Forward-time simulations under varying rates of migration for (A) $M = 0$ and (B) $M = 0.01$. The gray dots represent observed $F_{ST}$ and are connected by a line. The pink dots represent predictions from using our novel derivation of $B_{local}$ (Eq 2) to estimate $F_{ST}$ (Eq 4), the turquoise dots represent predictions of $B_{local}$ without accounting for the effects of migration (Eq 2; $m = 0$), and the dark blue dots are $F_{ST}$ from simulations run without selection (representing $F_{ST}$ under mutation-drift balance). The insets compare the simulations run with and without selection. Note the different scales for the x-axis on the main figures and insets. All simulations were run 750 times in a 10-deme island model with $N_{local} = 500$, $U = 7x10^{-3}$, $N_{local}t = 7.5$, $L_{selected} = 700$, $N_{local}m = \{0.5, 1.5, 5, 7.5, 10, 15, 25\}$, and for 250,000 generations.

that background selection can cause substantial reductions in $\pi_T$ (Fig 3), despite weak effects on $F_{ST}$. Under our parameter sets, this effect is most pronounced for lower strengths of selection against deleterious alleles under both low and high migration parameter sets, and it is weakly influenced by the introduction of low rates of recombination. Previous work has suggested that selective interference is more common in larger populations (reviewed by Ref. [51]); given that $N_{e, global}$ is on the order of $10^4$ in our simulations, this is likely the case. We investigated this further and found that this agrees with predictions under previous background selection theory derived for the *interference selection regime* [23]; this theory predicts a greater reduction in $B$ at lower strengths of selection for increasing population sizes (S1 Fig).

We find that the effects of background selection on $\pi_S$ and $\pi_T$ are dependent on the migration rate, which is to be expected given that migration influences $F_{ST}$ both directly and through its effects on background selection, and that $F_{ST}$ influences the effective population size of the metapopulation [27] (see *Methods Calculation of $F_{ST}$*). Under our low migration rate parameter set, we found that $\pi_S$ was reduced by a fractionally larger margin than $\pi_T$ (*cf.* Fig 3 to S2 Fig), resulting in the elevation of $F_{ST}$. However, in our high migration rate parameter set, $\pi_S$ and $\pi_T$ were similarly reduced for all simulated values of $N_{global}t$. For instance, we observed that, at $N_{global}t = 50$ with $m = 0.00045$ and $M = 0$, $\pi_T$ was reduced by a fraction of ~0.30 and $\pi_S$ by a fraction of ~0.45. At $N_{global}t = 50$ with $m = 0.09$ and $M = 0$, on the other hand, $\pi_T$ and $\pi_S$ were both reduced by a fraction of ~0.47. Overall, the results from our parameter sets suggest that background selection may cause substantial reductions in $\pi_T$ compared to neutrality, even despite having no effect on $F_{ST}$ (*cf.* Fig 1A and 1B to Fig 3A and 3B).

Moreover, we demonstrate that we are able to predict the effects of background selection on $\pi_T$ with reasonable accuracy under strong selection (Fig 3, $N_{global}t \geq 50$). Under lower

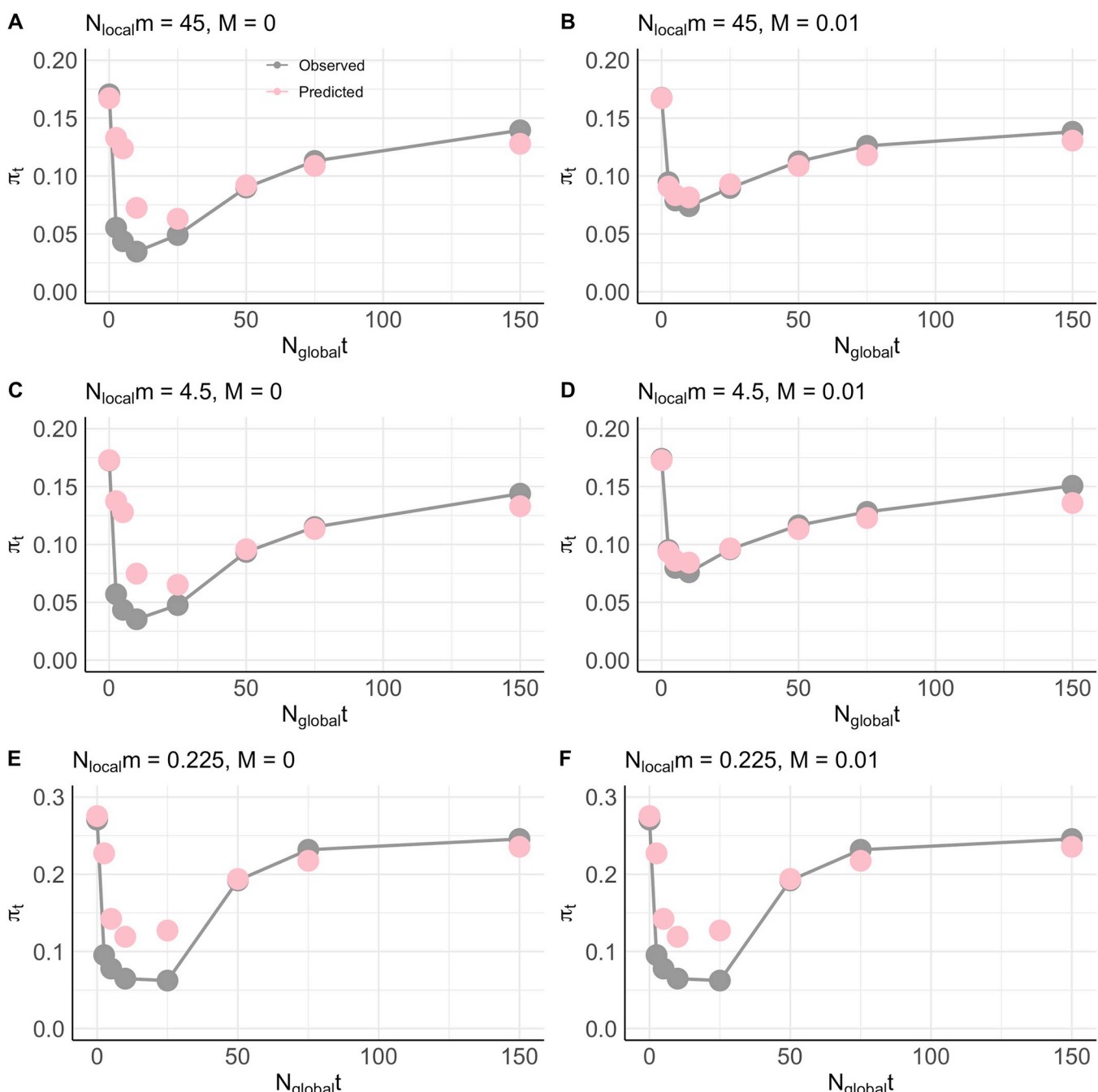

**Fig 3. Predicted $\pi_T$ versus observed $\pi_T$ in a metapopulation.** Forward-time simulations run with (A) $N_{local}m = 45$ and $M = 0$, (B) $N_{local}m = 45$ and $M = 0.01$, (C) $N_{local}m = 4.5$ and $M = 0$, (D) $N_{local}m = 4.5$ and $M = 0$, (E) $N_{local}m = 0.225$ and $M = 0$, and (F) $N_{local}m = 0.225$ and $M = 0.01$. The gray dots represent $\pi_T$ from forward-time simulations using a 10-deme island model with $N_{local} = 500$, $N_{local}t = \{0, 0.25, 0.5, 1, 2.5, 5, 7.5, 15\}$, $N_{local}m = \{45, 4.5, 0.225\}$, and are connected by a line. The pink dots represent theoretical predictions of total diversity (see *Methods*) using predictions of $B_{local}$ with incorporation of the migration effect (Eq 2) to estimate $G_{ST,}$ (Eq 5), $N_{e, global}$, and therefore $\pi_T$.

strengths of selection, $N_{global}t \leq 50$, however, our theoretical derivation overpredicts both $\pi_T$ and $\pi_S$ (S2 Fig). We considered that this may be either due to inaccuracy of the theory when interference is common, which is expected to be substantial under smaller $t$ in our parameter set, or an unconsidered artifact of population structure. To investigate this, we ran panmictic

simulations with $N = \{10^3, 10^4\}$ and $M = \{0, 0.01\}$ (S3 Fig), and found that the theory also poorly predicts $\pi$ in a panmictic population under smaller strengths of selection $t$ for when $N = 10^4$ and $M = 0$ (S3C Fig), parameters under which we expect the relatively greatest amounts of interference. On the other hand, the theory more accurately predicts $\pi$ under weaker strengths of selection $t$ for $N = 1000$ and $M = 0.01$ (S3D Fig), for which we expect the relatively lowest amounts of interference in our set of parameters. Thus, we have reasonable evidence to believe that inaccurate predictions of $\pi_T$ under $N_{global}t \leq 50$ are due to inaccuracy in the theory in accounting for the effects of interference, which has been previously noted [11].

For all of the simulations presented above, we also test predictions of the interference selection model of Good et al. [23], modified to consider the effects of migration (S1 Text). Overall, we find that the Good et al. [23] model performs similarly to the quantitative genetic model presented in the main text.

## Discussion

Prior to this work, models of background selection have been restricted to the case of populations that are panmictic [1,3–4,23–26] or have low migration rates [2,35–36]. This represents a substantial gap in the literature, as we expect many, if not most, natural populations to exhibit low to moderate spatial subdivision. Here, we have extended quantitative genetic models of background selection [11,25–26] to account for the effects of migration in an island model, showing that migration, in addition to selection and recombination, acts to end the cross-generational association between a neutral allele and linked deleterious alleles. A useful property of these models is that predictions for $B$ may be made under weak selection, by considering that weakly deleterious alleles have an appreciable rate of fixation via genetic drift. We tested the predictions of our novel derivation using simulations and found that it best predicts $F_{ST}$ and $\pi_T$ under varying strengths of selection, recombination rates, and migration rates. Our work provides a mechanistic explanation for the previously observed lack of effect of background selection on $F_{ST}$ under high migration rates [37], and highlights parameter space under which we expect background selection theory to be inaccurate.

Our work suggests that the relative importance of the 'migration effect' in influencing $B_{local}$ depends on the magnitude of the heterozygous selection coefficient, $t$, and the migration rate, $m$. In cases where $t >> m$, we expect the effect of migration to be relatively weak, as selection will likely act to remove deleterious variants prior to their emigration from a population. Under this relatively low migration regime, we expect predictions of background selection to be accurate even without consideration of the migration effect, given that migration is sufficiently infrequent. This is indeed what has been previously observed. For instance, Zeng & Corcoran [36] show that they are able to predict $F_{ST}$ well using classic background selection theory in simulations with $N_{local}m = \{0.05, 0.25\}$ and $N_{local}t = \{50, 15\}$. This is also aptly described by Charlesworth et al. [35], where it is stated that the effects of background selection on $F_{ST}$ are primarily through reductions in diversity, $\pi_S$, within populations; this conclusion was evidenced by simulations with $N_{local}m = 1$. In cases where migration rates are comparable to or stronger than the strength of selection against deleterious alleles, that is, $m \geq t$, we expect deleterious alleles to emigrate from local populations at an appreciable rate prior to their removal via selection. Under this regime of non-trivial migration rates, we expect the effects of background selection on $F_{ST}$ to be significantly attenuated, as we expect greater similarity in reductions in $\pi_S$ and $\pi_T$.

We now highlight some limitations of our study and avenues for future work. Our study only looked at an island model. However, the effects of background selection on $F_{ST}$ and $\pi_T$

remain unexplored for other population subdivision models, such as 'isolation-by-distance' models with varying dimensionality. We know that $F_{ST}$ interacts with global $N_e$ differently for non-island model demographics [27]. Additionally, the modulating effect of migration on the effect of background selection and $F_{ST}$ remains unexplored in these other population subdivision models. Thus, we caution against the application of our model to more complex demographic scenarios. We also use various other simplifying assumptions, including fixed selection coefficients and a simple genomic architecture. We did not explore the effects of varying deleterious selection coefficients on $B$, but it is possible that the combination of both weakly and moderately deleterious alleles may result in more complex dynamics of background selection. Furthermore, we do not modify our theory to be applied to a continuous distribution of selection coefficients. However, this problem is addressed by the work of Santiago & Caballero [26], which may be extended to our present model. Additionally, we note that our model is restricted to the case of semi-dominance of deleterious mutations. Under low recombination regions with substantial selective interference and recessivity of deleterious mutations, there is evidence that associative overdominance may act oppositely to background selection, instead leading to increased genetic diversity [52–53].

In summary, we have provided a novel theoretical derivation of the effects of background selection in an island model; this model accounts for the effects of migration providing an opportunity to lessen the effects of background selection by truncating cross-generational associations between neutral and deleterious alleles in their reproductive success, and presents a first step into the application of background selection theory in systems with more complex demography. We test the accuracy of the theory under various parameter sets, demonstrating that it predicts $F_{ST}$ accurately across a wide range of parameters and predicts $\pi_T$ well when we expect selective interference to be weak.

## Methods

### Simulations

Our primary objective was to study how well the approximations used in the theory of background selection allow the theory to make accurate predictions of $B$, and to identify parameter regimes where the theory is less accurate. Thus, although the parameter sets used for this study were biologically relevant, in each simulation run the selection coefficient was held fixed and our parameters were not representative of any one study system nor a particular gene model.

We ran simulations in SLiM v3.7.1 [47] using an island model consisting of 10 demes with a local diploid population size, $N_{local}$ = 500 individuals, for a total of $N_{global}$ = 5,000 individuals. To delineate the effects of background selection on $F_{ST}$ and $\pi_T$ under varying strengths of selection, we ran simulations under the following parameter set: $N_{local}t$ = {0, 0.25, 0.5, 1, 2.5, 5, 7.5, 15}, $\mu$ = $10^{-5}$, $U$ = $7 \times 10^{-3}$, $r$ = {0, $1.414 \times 10^{-5}$}, $M$ = {0, 0.01}, and $m$ = {0.00045, 0.009, 0.09}, $t$ is the selection coefficient against heterozygotes with a deleterious allele, $\mu$ is the deleterious mutation rate per base-pair, $U$ is the total gametic deleterious mutation rate, $r$ is the per base-pair recombination rate, $M$ is the total recombination rate (in Morgans), and $m$ is the total migration rate (the fraction of individuals in each deme that are replaced by immigrants from all other demes). To delineate the effects of varying migration rates on the effect of background selection on $F_{ST}$, we also ran simulations under the following parameter set: $N_{local}t$ = 7.5, $U$ = $7 \times 10^{-3}$, $M$ = {0, 0.01}, and $m$ = {0.001, 0.003, 0.01, 0.015, 0.02, 0.03, 0.05}.

Fitness effects were multiplicative, such that $w_i = (1-t)^k$, where $w_i$ is the survival probability of individual $i$, $t$ is the selection coefficient against heterozygotes, and $k$ is the number of deleterious alleles carried by the individual. The genomic region consisted of $L_{neutral}$ = $10^4$ neutral sites embedded in a region surrounded on each side by 350 sites subject to deleterious

mutation, adding up to a total of $L_{selected}$ = 700 selected sites. This is not intended to reflect the neutral mutation rate of a particular gene model. Rather, we chose to simulate a large neutral region to minimize measurement error in our estimation of $\pi$, allowing us to draw more accurate comparison to theoretical predictions.

Our simulated region is equivalent to a region consisting of a number of sites under selection upon which recombination occurred at total rate $M$, with a non-recombining neutral site embedded in the center, similar to genomic regions used in previous theoretic work (e.g. refs. [3, 23, 36]. Each simulation was run 750 times to derive 95% CIs and for 250,000 ($50N_{global}$) generations to achieve equilibrium. Data from the simulations were saved as tree sequence files, and neutral mutations were overlaid using *msprime*. Tree sequence data was analyzed using the *pyslim* and *tskit* packages in Python 3.9.12 to output $F_{ST}$, $\pi_S$, and $\pi_T$.

## Calculation of $F_{ST}$

$F_{ST}$ in the simulations was estimated as:

$$F_{ST} = 1 - \frac{\pi_S}{\pi_B}$$

[30,54], where $\pi_S$ is the average proportion of pairwise differences between haplotypes sampled within a deme at a neutral locus and $\pi_B$ is the average proportion of pairwise differences between haplotypes sampled between demes at a neutral locus. We filtered neutral alleles for which the minor allele frequency was less than 0.05 prior to calculating $F_{ST}$.

## Calculation of $B$

For the case where deleterious mutations behave deterministically (i.e. the *background selection regime*), Nordborg et al. [4] predict $B$:

$$B \approx \exp\left[-\sum_{i=1}^{L_{selected}} \frac{\mu_i t_i}{(r_i + t_i)^2}\right] \tag{6}$$

For a neutral mutation embedded in a large region of $L_{selected}$ sites subject to deleterious mutation with fixed $t$ and $r_i$ and being sufficiently small that it is additive over contiguous sites, we may use a continuous approximation for a region under selection, yielding the following formula:

$$B = \exp\left[-\frac{2U}{2t + M}\right] \tag{7}$$

[3], where $U$ is the gametic deleterious mutation rate for the region, $t$ is the heterozygous strength of selection against deleterious alleles, and $M = r_i L_{selected}$ is the total recombination rate. The above models assume that genetic drift is insignificant; thus, $B = B_{local}$ for Eq 6 and Eq 7.

To predict $\pi_{local}$ instead of $N_{f,local}$, (i.e. $B_{local} = \frac{\pi_{local}}{\pi_{0,local}}$, where $\pi_{local}$ is the geographically local estimate of diversity in a subpopulation experiencing background selection and $\pi_{0,local}$ is the neutral expectation) under our quantitative genetic model of background selection, we must consider that neutral alleles that contribute to diversity patterns arose at different times in the past, and have therefore experienced different degrees of the effects of background selection. An in-depth explanation may be found in ref. [26] but we describe this phenomenon in brief below.

Neutral mutations that have occurred further back in time are expected to have more strongly experienced reductions in $N_{f,local}$ due to greater accumulation of autocorrelation in variance in reproductive success with deleterious alleles. On the other hand, we expect newer neutral alleles to experience weaker reductions in $N_{f,\,local}$ due to the relatively short time they are likely to have been associated with deleterious alleles. Thus, it is incorrect to use the asymptotic linkage inflation factor, $Q^2_{\infty,i}$, to estimate $\pi_{local}$ at equilibrium. Instead, we must consider that $Q$ is a function of time, and that not all neutral mutations in association with deleterious alleles have arrived at their asymptotic limit, $Q^2_{\infty,i}$, with regards to the effects of background selection. Once we have accounted for variation in the time of linkage between existing neutral and deleterious variants, we are able to estimate the relevant effective population size for estimates of $\pi_{local}$, termed the local heterozygosity-effective population size $N_{eH,local}$ [26].

The methodology is as follows. First, we find that $Q_i^2(\tau) = \left[\frac{1-(Z_i(1-r_i)(1-m))^\tau}{1-Z_i(1-r_i)(1-m)}\right]^2$. Then, we estimate the local fitness-effective population size of neutral sites that arose $\tau$ generations in the past: $N_{f,local,i}(\tau) = N\exp\left[-\sum_{i=1}^{S} V_{A,i}\frac{Q_i^2(\tau)}{2}\right]$, for neutral sites arising from $\tau = 0$ to $\tau = 10 N_{local}$ generations in the past; this is somewhat arbitrary, but we follow the convention of [11]. Then, we are able to estimate $B_{local}$ to predict $\pi_{local}$ as $B_{local} = \frac{\pi_{local}}{\pi_{0,local}} = \sum_{t=0}^{\infty}[\prod_{i=0}^{\tau=10N_{local}}(1-\frac{1}{N_{f,local,,i}})]/2N_{local}$ [11,26]. Given that we are considering variation in the number of generations of autocorrelation between neutral and deleterious alleles instead of using the asymptotic limit, this quantity is greater than the asymptotic fitness-effective population size, $N_{f,local}$.

It has been noted that the use of the asymptotic linkage factor, $Q^2_\infty$ (and therefore $N_f$), to predict diversity is reasonable under most human parameters [11], but we expect the deviation in prediction of $B_{local}$ using $Q^2_\infty$ to be greatest under weak selection and tight linkage [26], a common feature of our simulations. These are parameters under which we expect the longest cross-generation associations between neutral and deleterious alleles, as there is a lower rate of association-ending events. Thus, we opt for the more precise estimate of $N_{eH}$ for the purposes of our predictions of $\pi$ for our simulations.

## Calculation and prediction of $\pi_T$

In order to calculate $\pi_T$ in a metapopulation, we must consider its global effective population size, $N_{e,\,global}$. $N_{e,\,global}$ is directly influenced by the degree of population structure, and, in the case of an island model, may be estimated as $N_{e,global} = \frac{N_{global}}{(1-G_{ST})}$ [27,55]. $G_{ST}$ is the appropriate measure of population structure here, because it is standardized by total diversity due to its definitional use of $\pi_T$ as required by the theory [27]. For our predictions of $\pi_T$ under background selection, we calculated $G_{ST,\,predicted}$ using background selection theory (see *S1 Appendix*) and then computed $N_{e,\,global}$. Using $N_{e,\,global}$, we were then able to calculate the global effect of background selection, $B_{global}$, on $\pi_T$ in the metapopulation. Then, we calculated predicted total diversity $\pi_{T,\,pred}$ using the *tskit* package in Python 3.9.12 as:

$$\pi_{t,pred} = \frac{4N_{e,global}B_{global}\mu}{1 + 4N_{e,global}B_{global}\mu}$$

[56].

## Supporting information

**S1 Appendix. $F_{ST}$ with a finite number of demes and migration immediately preceding measurement.**
(PDF)

**S1 Text. Derivation of $B$ with migration effect and application to the Good et al. [23] method.**
(PDF)

**S1 Fig. Predicted $B$ in a panmictic population according to the Good et al. [23] method.** $B$ was computed for haploid deleterious selection coefficients of $s = \{0.0005, 0.001, 0.002, 0.005, 0.01, 0.015, 0.03\}$, haploid population sizes of $N = \{100, 1000, 10000, 100000\}$, $U = 7\text{x}10^{-3}$, and $M = 0$. Here we see that, as $N$ increases, the theory predicts greater reductions in $B$ for weaker selection coefficients ($s \leq 0.005$).
(TIFF)

**S2 Fig. Predicted $\pi_S$ versus observed $\pi_S$ in a metapopulation.** Forward-time simulations run with (A) $N_{local}m = 45$ and $M = 0$, (B) $N_{local}m = 45$ and $M = 0.01$, (C) $N_{local}m = 4.5$ and $M = 0$, (D) $N_{local}m = 4.5$ and $M = 0$, (E) $N_{local}m = 0.225$ and $M = 0$, and (F) $N_{local}m = 0.225$ and $M = 0.01$. The gray dots represent $\pi_S$ from forward-time simulations using a 10-deme island model with $N_{local} = 500$, $N_{local}t = \{0, 0.25, 0.5, 1, 2.5, 5, 7.5, 15\}$, $m = \{5\text{x}10^{-5}, 1\text{x}10^{-2}\}$, and are connected by a line. The pink dots represent theoretical predictions of total diversity (see *Methods*) using predictions of $B_{local}$ with incorporation of the migration effect (Eq 2) to estimate $G_{ST}$, (Eq 5), $N_{e, global}$, and therefore $\pi_S$.
(TIFF)

**S3 Fig. Predictions for $\pi$ in panmictic populations from Buffalo & Kern [11] and Hudson and Kaplan [3].** Forward-time simulations of a panmictic population run with (A) $N = 1000$ and $M = 0$, (B) $N = 1000$ and $M = 0.01$, (C) $N = 10\,000$ and $M = 0$, (D) $N = 10\,000$ and $M = 0.01$, The gray dots represent observed genetic diversity, $\pi$, from our panmictic simulations. The pink dots represent predicted $\pi$ using the model of Buffalo & Kern (2024) (Eq 2), equivalent to our model when $m = 0$, and the purple dots represent predicted $\pi$ using classic B (Eq 7).
(TIFF)

## Acknowledgments

We thank Tom Booker, Tyler Kent, Brian Charlesworth, and Tianlin Duan for helpful discussions and comments. Simulations for this study were conducted using the Zoology Computing Cluster supported by the Zoology Computing Unit at the University of British Columbia.

## Author Contributions

**Conceptualization:** Asad Hasan, Michael C. Whitlock.

**Data curation:** Asad Hasan.

**Formal analysis:** Asad Hasan.

**Funding acquisition:** Michael C. Whitlock.

**Investigation:** Asad Hasan.

**Methodology:** Asad Hasan, Michael C. Whitlock.

**Project administration:** Michael C. Whitlock.

**Resources:** Michael C. Whitlock.

**Software:** Asad Hasan.

**Supervision:** Michael C. Whitlock.

**Validation:** Asad Hasan, Michael C. Whitlock.

**Visualization:** Asad Hasan.

**Writing – original draft:** Asad Hasan, Michael C. Whitlock.

**Writing – review & editing:** Asad Hasan, Michael C. Whitlock.

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
