## [Decision Letter · Decision Letter 0]

21 Apr 2024

Dear Dr Hasan,

Thank you very much for submitting your Research Article entitled 'FST and genetic diversity in an island model with background selection' to PLOS Genetics.

The manuscript was fully evaluated at the editorial level and by independent peer reviewers. The reviewers appreciated the attention to an important problem, but raised some substantial concerns about the current manuscript. Based on the reviews, we will not be able to accept this version of the manuscript, but we would be willing to review a much-revised version. We cannot, of course, promise publication at that time.

You will see that all reviewers commend the manuscript for reporting a solid theoretical advance. They (and I) agree that extending predictions under background selection to subdivided populations is an important goal. In contrast, two of the reviewers are skeptical about the broader significance of the work. This skepticism, which I share, stems primarily from three issues: (1) the manuscript reports no comparison of model predictions to empirical data, (2) assumed parameter values may not be realistic, and (3) the effects of background selection tend to be weak (especially on Fst). To be reconsidered for publication, these concerns would need to be answered convincingly. I would understand if you decide to submit to a journal where connections to real data are less salient than PLoS Genetics.

If you decide to revise the manuscript for further consideration at PLOS Genetics, please aim to resubmit within the next 60 days, unless it will take extra time to address the concerns of the reviewers, in which case we would appreciate an expected resubmission date by email to plosgenetics@plos.org.

We are sorry that we cannot be more positive about your manuscript at this stage. Please do not hesitate to contact us if you have any concerns or questions.

Yours sincerely,

Bret Payseur

Academic Editor

PLOS Genetics

Justin Fay

Section Editor

PLOS Genetics

Reviewer's Responses to Questions

**Comments to the Authors:**

Reviewer #1: In this work, Hasan and Whitlock derive theory for the reduction in effective population size due to background selection (BGS) in a metapopulation model under both strong and weak BGS. The authors do this by finding the appropriate rate parameter for the Good et al. (2014) model under migration (their equation 8). Using forward simulations, they show their work has a close correspondence with their theory.

While I think this is nice theoretical work and helps us refine our expectations of the impact of BGS in more complex situations with migration, the paper would be much stronger with at least one empirical application.

## Major Comments

- Throughout the paper, the authors frame a lot of the discussion of BGS in terms of fitness variance. Recent work in quantitative genetic approaches to BGS (Santiago and Caballero, 2016 and Buffalo and Kern, 2024) frame things similarly and provides theory to predict reductions in diversity in terms of fitness variance. These models are arguably easier to build intuition around than the Good et al. (2014) method. Since this paper is trying to incorporate weak selection BGS theory with migration, it would be strengthened by trying to frame their theory in terms of these more intuitive models that seem to work quite well for Eukaryotic genomes (though they have remaining issues when 2Ns ~ 1). Is there a way to frame equations 2-3 in terms of the Santiago and Caballero model? I would also consider citing this recent work on weak BGS since it is quite pertinent to the present study.

- As mentioned, this paper would be greatly strengthened with some application to an empirical dataset. Does the new theory fit any data better than previous work? Are any conclusions drawn from previous work wrong under the new model?

- Line 86-87: There are many features (e.g. SFS) not captured by a simple rescaling of Ne under background selection models. I would include a sentence making this point, explaining that distortions in genealogies and thus the site frequency spectrum occur, and cite Good et al. (2014), Cvijović et al. (2018), and Good and Desai (2013). Cvijović et al. were cited earlier, but this is an important point to make to readers, that the BGS process is more than rescaling Ne in reality, even though that approximation works well.

- Line 494: I found this section on Lewontin's Paradox speculative and inaccurate. While it is good to mention that the authors' theory leads to more accurate predictions of diversity and Fst under more complex demographic situations with migration, there is nothing in the present paper that indicates BGS will now lead to reductions in diversity over the several orders of magnitude necessary to explain Lewontin's Paradox. This issue of the scale of the problem was first made by Coop (2016), and later Buffalo (2021) quantified the relationship between π and Nc and explicitly tried to test whether BGS could explain the reductions, finding it could not. The reductions in the present paper, e.g. those in Figure 3 are at most ~ 0.05, but again, the observed shortfall in Buffalo (2021) is several orders of magnitude.

## Minor Comments

- Line 28: looks like a possible formatting error: π and T look like they are the same size.

- Line 84: I would also cite Santiago and Caballero (2016) here, which extends this line of work, handling the weak selection case that Good et al's (2014) paper mentioned.

- Lines 99-101: I would rework this paragraph's introduction, as two important points are being made: BGS affects structured populations and BGS is mediated by genomic organization. First, it is worth pointing out that it is not just recombination rate that mediates the effect of BGS across the genome, but also the placement of functionally conserved regions along the genome. I would add mention of this.

- Line 125: I would change the citation position here as U/t[1] could easily confuse readers as some mathematical notation.

- Line 138: I would also consider the work of Santiago and Caballero (2016) and Buffalo and Kern (2024) here. In these works, the weak selection issue is approximated using quantitative genetic models of BGS, with fitness variance rescaled according to the rate of fixation of weakly deleterious mutations.

- Line 145: Rather than say "multiplying" I would say "rescaling".

- Line 212: Is this the first mention of Blocal? It needs to be described more clearly. What makes it local, e.g. is recombination or mutation deme specific?

- Throughout, the authors should make sure "exp" is not formatted properly (not italicized). In LaTeX this would be "\\exp".

- I do not see any link to GitHub or another code repository website. It is imperative that all code, data, and analyses be accessible to reviewers and future researchers.

Reviewer #2: In this manuscript, the authors derive expressions for the relative levels of neutral nucleotide diversity in the presence of background selection (BGS) while accounting for migration when selection is weak and the population is likely under the interference selection regime. They do so by extending the previously used approach presented in the Appendix of Charlesworth 2012 (which had accounted for selection and recombination but assumed panmixia) to account for migration and obtain the relative rate of coalescence in the classical BGS regime. Then they use Good et al.’s approach to account for interference and predict FST values as well diversity in the metapopulation.

Overall this is an important contribution to the field and I strongly recommend the acceptance of the submitted manuscript. I do, however, have a few suggestions involving writing as I believe that the manuscript can be improved to reach a broader audience.

Major comments:

-The authors have done a good job of indicating what work was done previously and what new work was performed in this study. However, that has resulted in the paper being somewhat scattered. It was hard for me to follow what exactly was done and how. For instance, one reaches Figure 1 and it’s not clear at all what the different ways of calculating FST are. I would recommend clearly restating in the results section exactly how the green, yellow, and purple points were obtained.

-In your section “Derivation of B with ‘migration effect’ ”, it was easy to follow the derivation once I read through the Appendix of Charlesworth 2012. But it’s difficult to follow it independently. I would strongly recommend that the authors expand on the derivation a bit so that one does not have to read another paper to understand the derivation presented in the current study.

-Your equation on line 235 -> Can you please explain why your summation is not over [1 - t_i - r_i’ - m]^n , if we were to precisely follow the logic of the Appendix in Charlesworth 2012?

-It would be highly useful if the authors presented the migration rate in terms of N_local*m, instead of simply “m” (this applies to all main figures). It’s very hard to understand the context of these migration rates. Relatedly, it would be helpful if the authors can discuss their results on high vs low migration rates in the context of N_deme*m – what are high and low migration rates? For instance, your high migration cases have N_local*m = 45, which is extremely high, such that the population would probably be considered a panmictic one in practice.

-Please make publicly available the associated code and scripts used to generate the results in this paper.

Minor comments:

-Line 455-456 -> You say that deleterious alleles are typically expected to be weakly deleterious. That is not what most studies have inferred. At selected sites, we usually observe a large proportion (at least 50%) of moderately and strongly deleterious mutations. The statement needs to be supported by citations or other reasons.

Reviewer #3: See attached pdf.

**Have all data underlying the figures and results presented in the manuscript been provided?**

Reviewer #1: **No: **I could not find any link to code, etc. and have included this as a comment.

Reviewer #2: **No: **I don't see an associated github link.

Reviewer #3: Yes

PLOS authors have the option to publish the peer review history of their article (what does this mean?). If published, this will include your full peer review and any attached files.

Reviewer #1: No

Reviewer #2: No

Reviewer #3: No

---

## [Decision Letter · Decision Letter 1]

11 Sep 2024

Dear Dr Hasan,

Thank you very much for submitting your Research Article entitled 'FST and genetic diversity in an island model with background selection' to PLOS Genetics.

The manuscript was fully evaluated at the editorial level and by independent peer reviewers. The reviewers appreciated the attention to an important topic but identified some concerns that we ask you address in a revised manuscript.

Purely theoretical works are eligible for publication in PLoS Genetics, as long as they convincingly address a significant problem. On this front, I think you have succeeded. At this point, it is important to thoughtfully answer the remaining criticisms of Reviewer 3. More care is warranted in your description of the history of research on background selection and interference selection. There are still questions about the extent to which the simulation setup is realistic. This aspect needs further clarification, even if you elect not to change what is simulated.

We ask that you:

1) Provide a detailed list of your responses to the comments from Reviewer 3 and a description of the changes you have made in the manuscript.

To resubmit, log into your Editorial Manager account and select the option 'Revise Submission' in the 'Submissions Needing Revision' folder.

Yours sincerely,

Bret Payseur

Academic Editor

PLOS Genetics

Justin Fay

Section Editor

PLOS Genetics

Reviewer's Responses to Questions

**Comments to the Authors:**

Reviewer #1: I commend the authors on their nice improvements to the theory sections of the manuscript.

However, they rejected the AE and reviewers' suggestions to include an empirical application. This would not be a problem at other journals, but in checking the PLoS Genetics guidelines, it says manuscripts will be rejected if:

- Absence of substantive insight into biology or disease pathogenesis

- Incremental methods development studies that do not include an application to a biologic problem or system that reveals novel insight

Unfortunately, I have no option but to reject the manuscript due to these requirements. I hope to see this nice work published in another journal though.

Reviewer #2: The authors have done an excellent job of addressing all comments. This is a strong contribution to the field. I would however recommend improving the GitHub repository so that it is self-explanatory.

Reviewer #3: Review uploaded.

**Have all data underlying the figures and results presented in the manuscript been provided?**

Reviewer #1: Yes

Reviewer #2: Yes

Reviewer #3: Yes

PLOS authors have the option to publish the peer review history of their article (what does this mean?). If published, this will include your full peer review and any attached files.

Reviewer #1: No

Reviewer #2: No

Reviewer #3: No

---

## [Decision Letter · Decision Letter 2]

21 Oct 2024

Dear Dr Hasan,

We are pleased to inform you that your manuscript entitled "FST and genetic diversity in an island model with background selection" has been editorially accepted for publication in PLOS Genetics. Congratulations!

Yours sincerely,

Bret Payseur

Academic Editor

PLOS Genetics

Justin Fay

Section Editor

PLOS Genetics

Comments from the reviewers (if applicable):

Reviewer's Responses to Questions

**Comments to the Authors:**

Reviewer #3: I think the paper is now acceptable; the authors have responded to my comments.

**Have all data underlying the figures and results presented in the manuscript been provided?**

Reviewer #3: None

PLOS authors have the option to publish the peer review history of their article (what does this mean?). If published, this will include your full peer review and any attached files.

Reviewer #3: No

**Data Deposition**

http://datadryad.org/submit?journalID=pgenetics&manu=PGENETICS-D-24-00300R2

**Press Queries**

---

## [Editor Report · Acceptance letter]

21 Nov 2024

PGENETICS-D-24-00300R2 

FST and genetic diversity in an island model with background selection 

Dear Dr Hasan, 

We are pleased to inform you that your manuscript entitled "FST and genetic diversity in an island model with background selection" has been formally accepted for publication in PLOS Genetics! Your manuscript is now with our production department and you will be notified of the publication date in due course.

With kind regards,

Zsofia Freund

PLOS Genetics

On behalf of:
